# Policy-driven transformation of global solar PV supply chains and resulting impacts

Can Cui [1,2], Katherine Emma Lonergan [1,2] & Giovanni Sansavini [1,2] ✉

Tripling renewable energy capacity by 2030 requires increasing technology production capacity, including solar photovoltaics (PV). Current supply chains rely heavily on Chinese production; however, this situation is not aligned with regions aiming to increase self-sufficiency, decrease supply chain emissions, and increase local job opportunities. Here, we apply a supply chain optimization model to perform scenario analysis of the PV supply chain development through 2021-2030 considering various European economic and job creation goals. Irrespective of regional goals, we find that China is poised to remain a globally dominant supplier through 2030, especially in terms of lower-value PV components, given that future demand requires increasing global production capacity by a factor of at least 1.5. We find that some regional supply chain goals can be co-beneficial, for example in terms of joint job gains and increased regional self-sufficiency. However, pursuing highly isolationist policies can introduce cost-significant inefficiencies. Our results highlight that an open trade policy is key to minimizing costs, even when considering security and environmental supply chain objectives.

Achieving a net-zero energy transition relies heavily upon the deployment of solar photovoltaic (PV) systems. Countries have pinned much of their ambitions for renewables expansion on solar PV, with the technology expected to account for 50% of new renewable capacity and be the most important renewable technology in terms of total capacity by 2030[1]. Solar PV's appeal can be attributed to several factors, including the technology's cost-effectiveness, high social acceptance, and potential for global deployment[2,3].

Another factor in solar PV's popularity as a low-carbon technology is its market potential. The total value of global solar PV trade is valued at ~40 billion United States dollars (USD) per year—but to meet net-zero goals, the rate of annual solar PV additions must more than quadruple[4]. Chinese production accounts for over 80% of the current market[4], a position obtained with help from large government subsidies for research and development[5,6] and investments in primary production[7,8]. Lower-cost energy and labor support the cost-competitiveness of Chinese solar PV[9]. Besides supplying most of the world's demand for solar PV, China's support for solar PV has also helped drive down costs quickly, saving the global economy at least

USD 67 billion between 2008 and 2020[7]. The global benefits, past and future, of Chinese solar PV manufacturing capacity to global deployment are tightly linked to open trade policies, which facilitate global technology learning and making the most of existing capacities[7,10].

However, Chinese supply chain dominance is not viewed entirely favorably for several reasons. First, developing low-carbon technologies is an attractive and future-oriented market development opportunity. Countries are considering how they can also share financial, innovation, and job creation benefits, instead of concentrating on a single supplier. Solar PV's relatively assured importance to the transition, comparatively low reliance upon critical minerals[11], and comparatively low design complexity and customization requirements[12] help increase the attractiveness of building up solar PV manufacturing capacity compared to other low-carbon technologies. Moreover, since producing a solar PV panel involves several intermediate products—namely, polysilicon, ingots, wafers, cells, and modules[4]—there is an opportunity for countries to contribute to and benefit from at least some part of the supply chain. For example, countries in Southeast

[1]Institute of Energy and Process Engineering, ETH Zurich, 8092 Zurich, Switzerland. [2]Reliability and Risk Engineering, ETH Zurich, 8092 Zurich, Switzerland. ✉e-mail: sansavig@ethz.ch

Asia such as Malaysia, Thailand, and Vietnam have developed manufacturing capacity for module assembly[4].

Second, relying primarily on a single supplier can lead to a clear supply chain dependency. Such a dependency raises concerns for supply chain resilience and, in some cases, for geopolitical reasons. The transition to a low-carbon energy system holds the potential to reshape global energy markets and move away from existing fossil-fuel-based dependencies to more autonomous and diversified supply[13]; moving from dependency on one country to another is, therefore, undesirable. The desire to reduce trade with single countries may also stem from broader political shifts, such as within American federal politics[14].

Policymakers have two primary strategies for increasing self-sufficiency in the PV supply chain, namely, limiting the availability of foreign products and introducing supportive measures for local manufacturers. Both strategies for increasing PV self-supply have pros and cons. The appeal of trade policies, like tariffs, is that they protect local manufacturers; however, such policies can lead to price surges, hinder adoption rates, and impede technology learning[7,10,15,16]. Directly supporting local manufacturers is meanwhile appealing because subsidies, tax incentives, and grants for research and development can boost domestic production capabilities and foster innovation. However, these direct support measures require substantial upfront capital commitment, especially in the early stage of PV development[17]. In practice, both strategies are frequently used. For example, in May 2024, the United States (US) has proposed a 50% tariff on Chinese solar cells to protect the local industry[18]. On the supportive side, the US Inflation Reduction Act allocates 370 billion USD to energy security and climate spending, aiming to enhance domestic manufacturing, which had an impact on the decision-making of companies' relocation[19]. China also relied on a direct support approach to build its solar PV supply chain: government support included fiscal support[20], energy incentives[15], research and development (R&D) funding[5], tax rebates[20], land use incentives[15], and infrastructure investments[20].

In the European Union (EU), developing local PV manufacturing capacity fits into the bloc's wider strategic plans for developing its low-carbon economy[21,22]. Besides capitalizing on the potential financial benefits and supply chain independence, European policymakers hope that local low-carbon technology production can stimulate high-quality, low-carbon jobs[21] and reduce the environmental burden of the manufacturing process itself. The environmental benefits of producing solar PV in Europe come from Europe's lower-carbon energy supply[23,24], avoided emissions linked to transportation[23], and mitigating the risk of carbon leakage[25]. Building up the solar PV supply chain can also be seen as a chance for Europe to regain a missed opportunity: the initial wave of solar PV adoption was led by European demand and, for a time, Germany's manufacturing was a competitive supplier[4]. The EU has goals to reach 30 gigawatts (GW) of operational solar PV manufacturing and 40% self-production of net-zero technologies, including solar PV, by 2030[26]. Several policies are in place to achieve these goals, including providing production[27] and purchase subsidies[28], upgrading skills[29], implementing a carbon tariff on goods entering the EU[25], and supporting industry networks like the European Solar Photovoltaic Industry Alliance[30].

Despite the extensive and varied policy support, it is unclear whether European policymakers will be able to achieve their ambitions for localizing PV supply chains. Building up low-carbon manufacturing is a complex task, requiring a supportive regulatory environment[12] and access to skilled labor[31]. Supplier preferences[32] and cost differences[33] also influence the industry's willingness to alter supply chain patterns. Singular events and policies can disrupt global supply PV chains, as did the global COVID pandemic[15] and American trade policies targeting Chinese production[34]. Recent work suggests that relocating PV manufacturing outside of China could help reduce supply chain emissions[35], but that systemic financial support is required to

overcome financial barriers[23]. As such, the full scope of the opportunities, trade-offs, and impacts of European policy action is yet unclear.

To address these gaps, we examine how European policy actions aimed at building a local solar PV supply chain affect global trade flows and quantify the associated environmental and social impacts. We test different European policy scenarios using a bi-objective optimization model considering costs and job creation, and analyze the corresponding impacts over the period from 2021 to 2030 (Methods−Modeling framework). Relying on an optimization model allows us to test "what-if" scenarios associated with strong policy action (see Methods− Scenarios). We find that setting regional policy goals can help reshape and diversify the global PV supply chain; however, China remains a dominant supplier in all scenarios. We furthermore find that supply chain development goals can be cohesive, i.e., aiming to localize the European supply chain also supports local job growth and decarbonization. Our research contributes to the literature in three ways. First, we demonstrate the impact of regional policy action on future global supply chain networks, with and without coordinated action of partner regions. By doing so, we overcome the limitations of past work that takes only a retrospective perspective[7,15] or focus only on local policy impacts without considering knock-on supply chain effects[33,35]. Second, our results reinforce existing literature in suggesting that open trade policy is key to minimizing costs and creating jobs, even when considering global and local security and environmental impacts. Finally, we provide an in-depth analysis of the specific trade-offs facing European policymakers, thereby providing region-specific decision-making input that has hereto been lacking.

## Results

To analyze the global PV supply chain and identify potential transformations, we develop an optimization-based supply chain model for PV production (Methods). This model follows the real-world trend of cost-driven decision-making in production and trade, while also supporting policies that promote local supply and job creation. The model encompasses 12 regions to provide a comprehensive view of the global solar PV market. We consider five steps of PV production: polysilicon, ingots, wafers, cells, and modules (Supplementary Fig. 1)[4]. Figure 1a shows the schematics of the model based on the demand and manufacturing capacity data in 2021. To begin, each region is assigned product manufacturing capacity based on industry reports and statistics[4,36]. For example, Germany can produce products other than ingots, while China owns the manufacturing capacity for all five products. To satisfy the regional demand for PV modules, the regions can either acquire PV via trade or produce components locally, shown as the dashed lines linking the products among regions in Fig. 1a. Each region's demand for PV modules changes over time, and they can expand the manufacturing capacity and store products, if needed (Fig. 1b). The model covers the global PV supply chain from 2021 to 2030, using weighted multi-objective optimization for total industry costs and job creation (details in Methods: Modeling framework) to analyze the impact and pathways of the supply chain transition. Figure 1c shows the module production in 2030 relative to 2022, considering the dual goals of minimizing industry costs and maximizing job creation, indicating a common increase in local production across the regions. For example, module production in China and the US can become 1.5 times and 3.8 times the level in 2022, respectively; while for European regions (EUR), the production is modeled to become over 6.5 times the level in 2022. This result is roughly in line with the estimates provided by the International Energy Agency[37].

### Potential global PV supply chains in 2030

We investigate the structure of potential global PV supply chains in 2030 considering different European development priorities. We consider five scenarios with different sets of objectives and constraints (Methods and Data: Scenarios). In the baseline scenario, the model

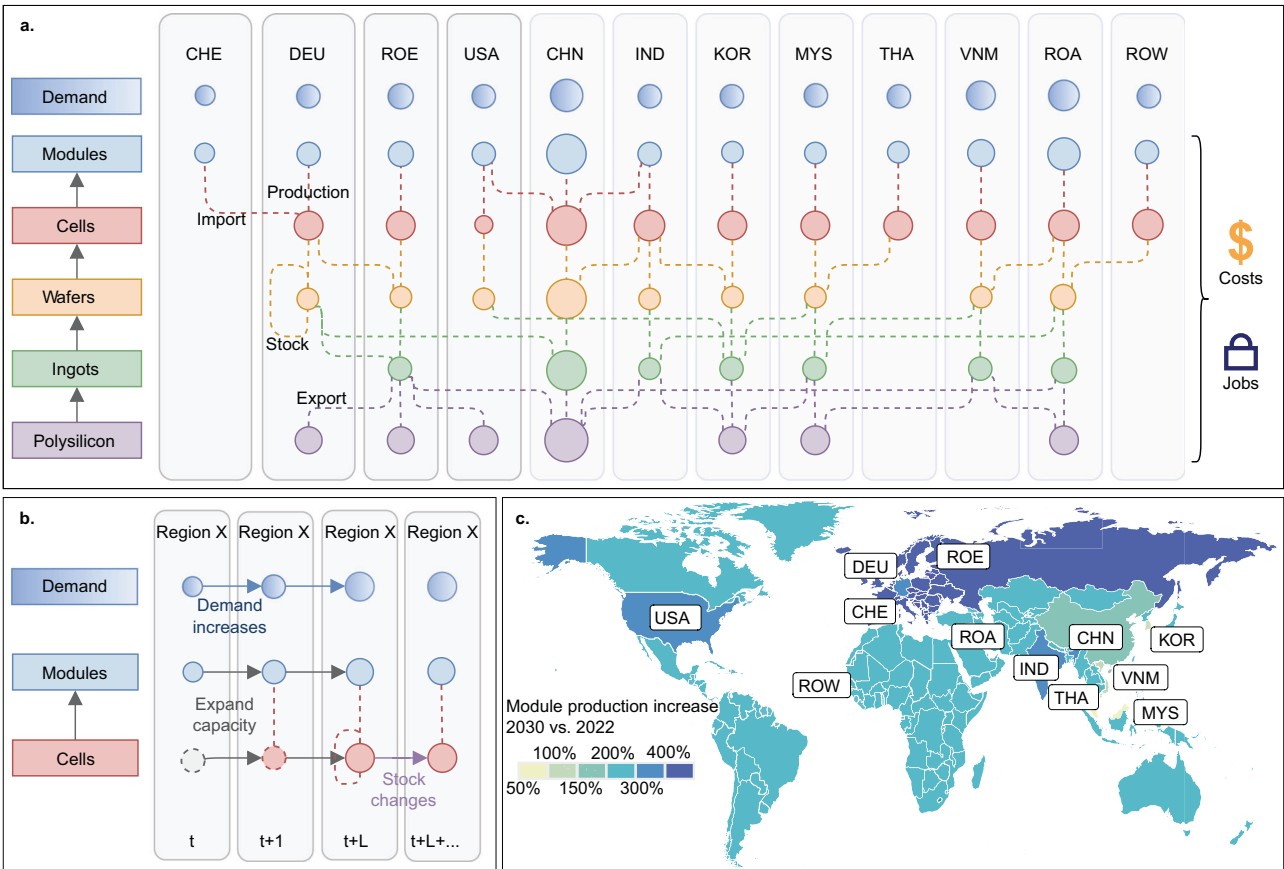

**Fig. 1 | Schematics of the model and definitions of key elements, basic processes, and regions.** Panel **a** shows the model framework with 12 regions participating in five components' supply, based on the demand and manufacturing capacity data in 2021. Point size shows the relative module demand and production capacity. Panel **b** shows the options for meeting increasing demand within a region, with time series from year t to several years' lead time (L) beyond year t. The process is shown for a sample region. Panel **c** shows the relative increase in module production from 2022 to 2030 for the modeled baseline scenario.

finds supply chains that achieve the lowest industry costs while maximizing job creation objectives over the 2021–2030 period, without considering European strategic interests. In the second scenario, the model finds supply chains that are low-cost and that maximize European job creation (MEJ scenario), while in the third scenario, Europe is set to achieve at least 40% self-sufficiency with a focus on maximizing job creation[26]. The fourth and fifth scenarios consider additional European policy intervention, respectively considering that European regions stop importing from China starting in 2024 and that European regions subsidize module production by 20%. Throughout, we assign the costs of building up manufacturing capacity to industry, while government subsidies are excluded from the industry costs.

Figure 2 shows the trade flows of the five solar PV component products in 2022 and the five scenarios' results in 2030. In 2022, mainland China produced over 81% of the products, with the share for ingots and wafers reaching 98% (Fig. 2; purple arrows). Compared to other components, the supply of PV modules is the most diverse in terms of both the number of suppliers and total market share. While polysilicon and modules see more diverse global production than other products, China's low production costs–up to 35% lower than in Europe[4]–make the country the most cost-competitive, attracting worldwide buyers. The results of the optimization model suggest that the key product flows resemble the current supply chain, but there are still opportunities for the supply chain transition to be more cost-effective and job-supporting (Supplementary Fig. 2).

China retains a dominant market position into 2030 (Fig. 2b), hosting as much as ten times manufacturing capacity as the second largest supplier (Supplementary Fig. 3). However, compared to the 2022 market, the 2030 scenario results suggest that building a supply chain aiming to minimize industry costs and maximize job creation would lead to a more diverse manufacturing market, as the suppliers participate with higher market shares in 2030. All scenarios show a trend towards reduced reliance on imports and increased self-sufficiency. In all scenarios, Europe is completely self-sufficient in module production, due to the relatively low cost of building manufacturing capacity and the goal of maximizing jobs. Europe is also able to achieve its 40% self-sufficiency targets by 2030 for all PV products in three different scenarios: when maximizing European jobs while being forced to meet the self-sufficiency goals, when introducing trade barriers against China, and when providing subsidies for European PV modules (MEJ + Europe >40%, European trade barrier, and European module subsidies in Fig. 2b). Other regions also increase their market shares. For example, the Rest of Asia (ROA) becomes more important in supplying polysilicon, ingots, and wafers across all 2030 scenarios.

Increasing European self-sufficiency and job creation drive the supply chain's relocation to Europe (blue arrows in Fig. 2; see scenarios MEJ and MEJ+ Europe >40% self-supply). This effect is strongest for upstream components, as Europe is more self-sufficient in downstream products to begin with. Imposing a European trade barrier with China shifts the Europe's dependence from China to Malaysia, thus not totally achieving the goal of increased self-sufficiency but merely transferring dependency to another country. Similar effects can be expected if both the USA and Europe implement trade barriers against China–the unmet demand turns to India, Malaysia, Vietnam, and ROA

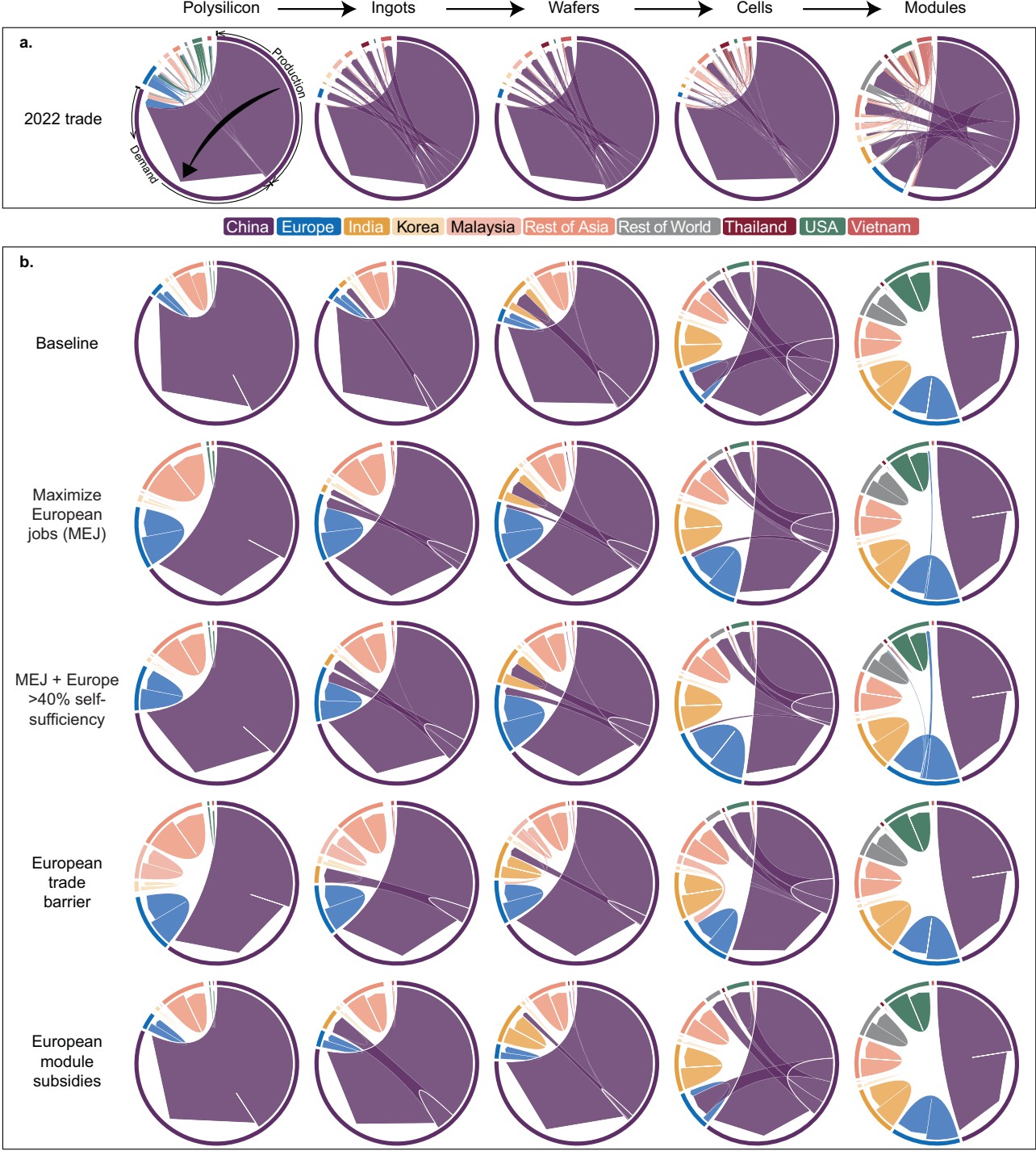

**Fig. 2 | Overview of global PV supply chains as shown by region-to-region trade flows for individual PV products.** Outer rings compound production and demand for the ten indicated regions. Arrows indicate regional trade flows and show the share of supply and demand for each outer ring; for example, the black arrow in (**a**) emphasizes that the majority of China's polysilicon supply is used to meet the domestic national demand and the remaining demand is mainly supplied from Europe and Malaysia. **a** shows 2022 trade flows and **b** shows the 2030 trade flows according to the tested scenarios. Trade flows for individual products are shown left-to-right following the PV value chain, with modules being the products sold to end-consumers.

(Supplementary Fig. 4). Banning products along the entire supply chain has greater impacts in rerouting supply chains than banning individual products (Supplementary Fig. 4). Moreover, banning the trade of downstream products simply shifts the dependence of upstream products on other regions, i.e., banning module imports leads to more imports of cells (Supplementary Fig. 4). Subsidizing module production in Europe has little impact on upstream products,

as Europe is already highly self-sufficient in module production, even in the baseline scenario for 2030.

Increasing market shares requires regions to increase their production across the PV supply chain. For example, China needs to increase its production of polysilicon, ingots, wafers, and cells; ROA needs to expand its production of polysilicon, ingots, and wafers; Europe requires growth in ingots, wafers, cells, and modules; and India

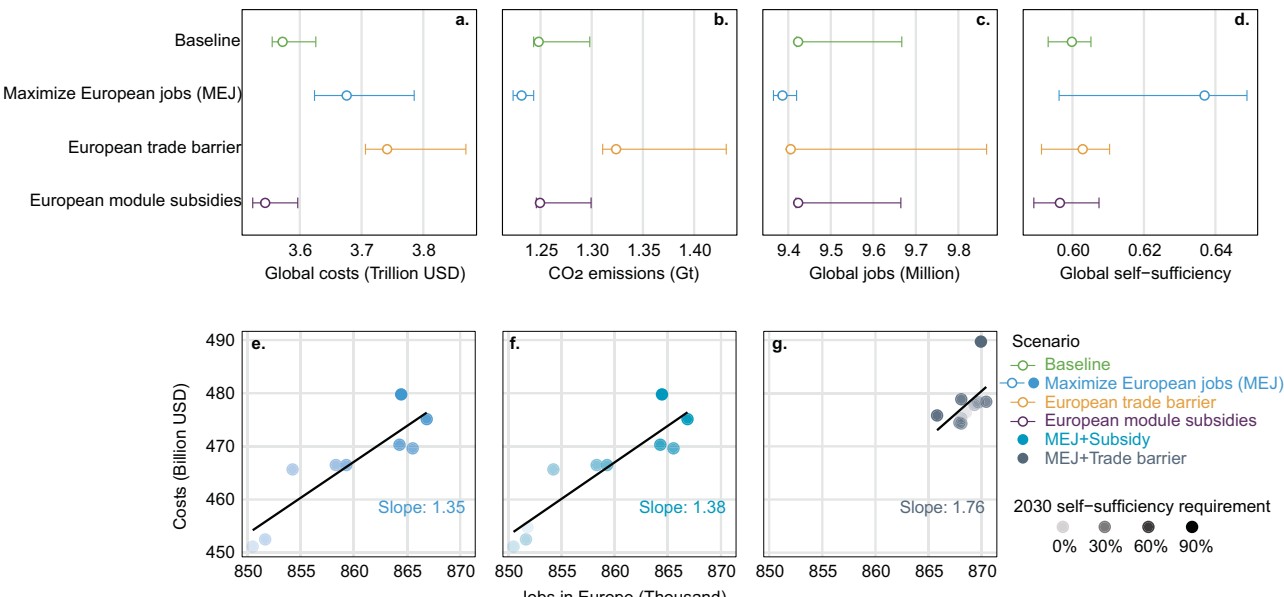

**Fig. 3 | The cost and impacts of European policy actions on PV supply chains.** Panels **a–d** show the scenario-by-scenario results on global **a** cumulative costs, **b** cumulative emissions, **c** cumulative job creation, and **d** average self-sufficiency across years. The circles show the median and the error bars show the min-max range. Panels **e–g** show the cumulative cost-jobs trade-offs according to policy actions aiming to **e** maximize European jobs (MEJ), **f** maximize European jobs and provide a production subsidy, and **g** maximize European jobs and impose a trade barrier on all Chinese PV products. The point shading in panels **e–g** corresponds to different European self-sufficiency targets, which are modeled as increasingly stringent optimization constraints.

needs to focus on increasing its cell production (Supplementary Fig. 3). One risk that emerges is the potential of oversupply in PV production capacity, already anticipated (Supplementary Fig. 5)[38]. The global efficiency of manufacturing capacity, measured as the share of actual production relative to total capacity, falls in the range of 58–79% (Supplementary Fig. 6). Even lower efficiencies are observed for specific national-level production cases, such as for module production in Malaysia and Vietnam and wafer production in Germany and the USA, with less than 10% efficiency. In contrast, the efficiency of ingots capacity exceeds 96% in ROE, India, and ROA (Supplementary Fig. 6). This uneven efficiency reveals regional imbalances since regions prioritize low industry costs and local supply; this results in under-utilized capacity.

**Global and regional performance of supply chains by scenario**

Pursuing regional supply chain goals can clearly affect global supply chains (Fig. 2) and, by extension, supply chain impacts in terms of cost, carbon dioxide emissions, job creation, and regional self-sufficiency. We next investigate how these four impacts differ according to policy scenario and the associated tradeoffs.

Figure 3a–d compare the global industry costs, carbon dioxide emissions, jobs, and average regional self-sufficiency across four scenarios. These scenarios consider factors such as job creation location, trade policies, and subsidy policies without imposing additional self-supply constraints. Adopting a Euro-centric focus for building global PV supply chains has clear effects on all four impact categories. For example, aiming to maximize European job creation provides 44.5% more jobs in Europe (but 0.4% less jobs globally), reduces global emissions by 1.6%, and increases global self-sufficiency by 4%. Likewise, if Europe stops importing PV products from China, total industry costs and emissions increase by about 5%, global self-sufficiency increases by 0.3%, and global job creation decreases by 0.1%. Finally, government subsidies in Europe, amounting to 7.1% of the baseline industry costs, help share the financial burden on European industry (0.8% globally) but lower the global average self-sufficiency by 0.3%. This effect occurs because localizing European production reduces dependence on other

regions and even reduces the need for other regions to build up their manufacturing capacity. In turn, this also reduces the ability of other regions to meet their local demand with local supply. These results collectively indicate the power of regional (e.g., European) policies on global supply chains and their resulting global socioeconomic and environmental impacts. From the perspective of components, poly-silicon is the most expensive product, while module supply creates the most jobs with the highest self-sufficiency (Supplementary Fig. 7).

Manufacturing location entails critical social and environmental trade-offs, particularly between carbon emissions and job creation, which are heavily influenced by regional differences in industrial production processes. Manufacturers in developed regions, such as Europe, benefit from lower-carbon energy sources but generate fewer jobs due to higher labor costs[4] (see Supplementary Data 1). This results in the scenario where maximizing European jobs leads to the lowest carbon emissions but also the least global job creation (Fig. 3b, c). In contrast, trade barriers between Europe and China shift production to other Asian regions with even higher carbon intensity and more variable labor conditions than in China, increasing global $CO_2$ emissions while only marginally reducing job creation (by 0.1%). These results highlight a key tension: policies favoring localized production in developed economies may reduce emissions but limit global employment opportunities, whereas shifting supply chains to lower-cost regions can expand job creation at the expense of higher environmental impact.

Considering the scenarios aiming to maximize European jobs in more detail reveals the cost-job trade-offs. The scatter plots shown in Fig. 3e–g summarize the total jobs created versus total European PV supply chain industry costs from 2021 to 2030. In all three scenarios, achieving more jobs and higher self-sufficiency in Europe correlate with higher cost. Between 2021 and 2030, transforming the PV supply chain could create 850–870 thousand full-time manufacturing jobs in Europe; this represents an increase of 45-48% compared to the number of European jobs in the baseline scenario. Correspondingly, the cumulative industry costs increase by 94–119 billion USD compared to the baseline scenario. Trade disruption leads to more job creation but

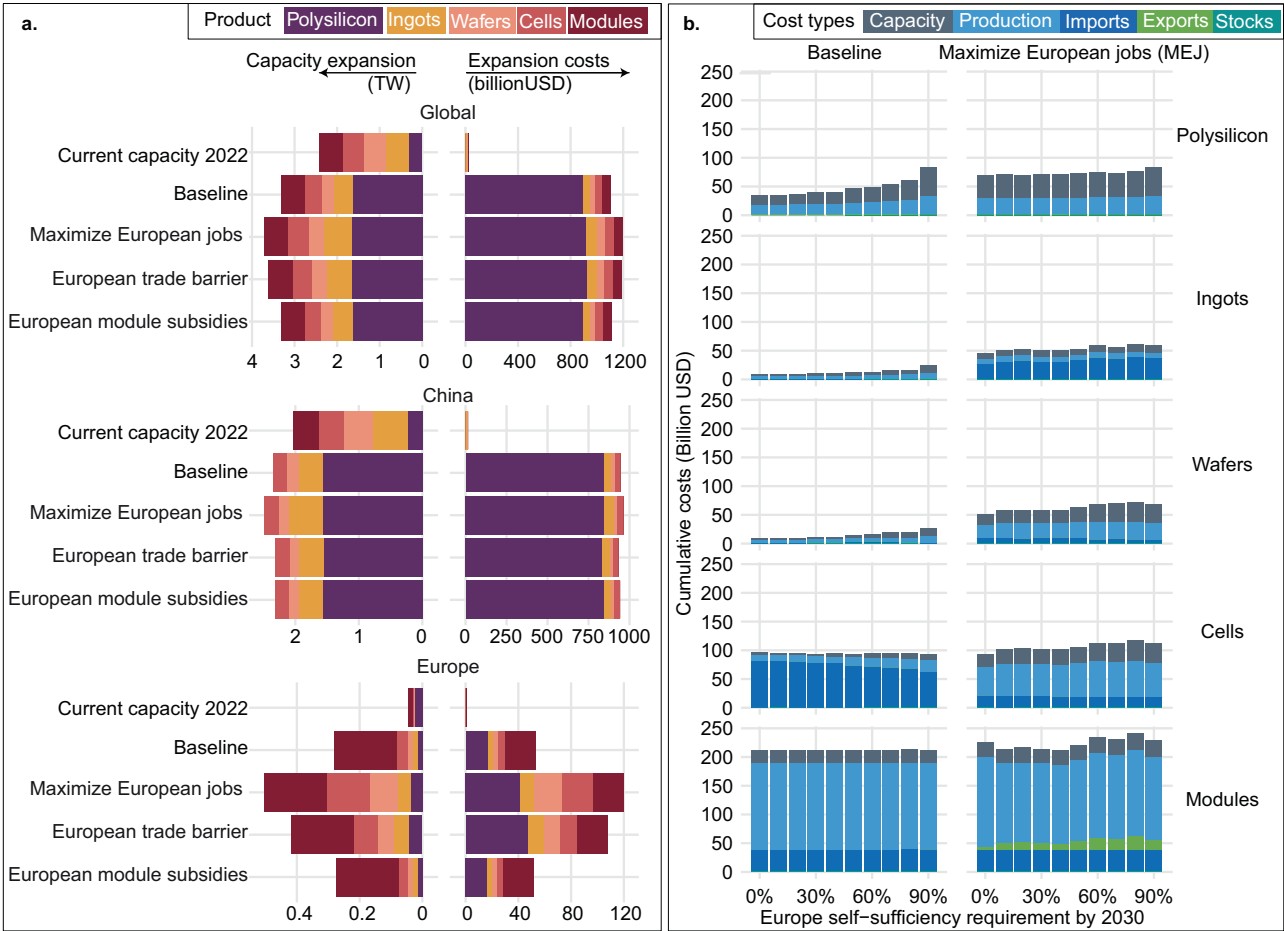

**Fig. 4 | Capacity and cumulative expenses of PV supply chain transformation.** **a** shows capacity expansions and capacity expansion costs for global total, China, and Europe, for the year 2022 as reference (labeled as "Current capacity 2022"), and for cumulative from 2023 to 2030 in the four scenarios, by product in colors. **b** shows Europe cumulative industry costs and self-sufficiency requirements for each product by 2030, by scenario and cost type in colors.

at a higher cost: creating 1000 more jobs in Europe requires 1.76 billion USD, which is about 30% higher costs than the other two scenarios. Overall, these results suggest that strictly aiming to avoid imports and increase jobs without considering other supply chain impacts would be a costly strategy for European policymakers.

**Transformation requirements and costs**
Governments can stimulate local PV production to reduce the dependence on foreign supply by investing in manufacturing capacity and subsidizing production. However, significant capacity expansion and associated costs are required to achieve those goals. Figure 4a shows the global capacity expansion and corresponding expansion costs needed to bridge the gap between 2022 and 2030 for the scenarios considered. According to the baseline scenario, global manufacturing capacity in 2030 must increase by over 3.4 TW (over 1.5 times) compared to 2022 values. In Europe, a 6.1-fold capacity increase is needed compared to 2022 values for the baseline scenario and an 11-fold capacity increase for the "Maximize European jobs" scenario, focusing on expanding module, cell, and wafer production. Within Europe, Germany is expected to build 20% of the capacity, while ROE is expected to expand up to 80% of the manufacturing capacity especially for products other than modules (Supplementary Fig. 8). Chinese manufacturing capacity must increase by around 1.3 times its 2022 capacity to supply domestic and other regions' demand, especially for polysilicon.

The highest capacity increases occur in scenarios that maximize European jobs, showing that regional goals can introduce significant

trade-offs to a globally efficient energy transition. Similarly to the results for local impacts of the transformation, we find that job creation objectives and trade disruptions are linked to higher capacity expansion and industry costs, while subsidies in Europe incur capacity expansion and costs to industry compared to other policy scenarios. The cost for global capacity expansion across all scenarios is over 1 trillion USD (Fig. 4a); more than 900 billion of this is set to be invested in China, which is nearly three times the amount the country invested in solar manufacturing and power generation in 2023[39]. We estimate that capital costs for European supply chain development to be in the range of 50–120 billion USD. Our cost estimates for the baseline scenario align with those suggested by the European Solar Industry Alliance[40] in terms of the financing required to increase downstream wafer, cell, and module capacity. However, our financing estimates for developing a fully European PV value chain imply far greater efforts than currently called for in other proposals[9], namely, developing polysilicon and ingot manufacturing requires an additional 20–52 billion USD.

Capacity expansion costs account for only 12.4% (as median) of the total industry costs, and the actions needed for supply chain transformation are more than building sufficient capacity for local manufacturing. The cumulative industry costs include costs for capacity expansion, production, trade, and managing surplus stock. Figure 4b shows cumulative industry costs and self-sufficiency constraints for the baseline and the maximum European job creation scenario. In both cases, increasing the requirements for European self-sufficiency increases the supply costs for polysilicon, ingots,

wafers, and cells. The cost increase is primarily due to Europe's relatively higher industry costs in capacity expansion, production, and import expenses[41–43]. Production costs remain the predominant expense for modules. When compared to the global cost-and-job-optimal scenario (baseline case), pursuing increased employment in Europe (MEJ) leads to a nearly one-third (32.2%) rise in cumulative industry costs for the European PV industry. Maximizing European job creation also indirectly contributes to increased self-sufficiency, providing 14% higher self-sufficiency in module production. Additional self-sufficiency requirements on European PV only have a mild cost impact on the local industry: increasing self-sufficiency constraints from 10 to 90% only increases total costs to European industry by 6.3%. The same trend holds when considering additional trade barriers and subsidies (see Supplementary Figs. 9, 10).

## Discussion

Developing local manufacturing capacity in low-carbon technologies, including solar PV, offers the potential benefits of supporting self-sufficiency, job creation, and lower environmental impacts. Here, we show that regional European policy goals targeting these supply chain objectives can significantly reshape supply chain dynamics. Moreover, we find that achieving self-sufficiency and job creation goals could increase industry costs by up to 34% higher than those of a globally optimized supply chain. However, irrespective of European regional goals, China will maintain a predominant role in the solar PV supply chain due to the advantages of manufacturing capacity and costs, and the need to expand global capacity by over 1.5 times. As such, the goal for diversifying the global supply chain can only be partially fulfilled, but still presents notable benefits to countries and regions that apply concerted efforts to build up their local capacities.

Our findings agree with existing work that an open trade policy is key to achieving a low-cost energy transition[7,10]. Implementing trade barriers can raise the cost of job creation by up to 30% and increase global emissions by shifting production to countries with low production costs and carbon-intensive energy systems. Moreover, trade barriers may simply shift rather than resolve trade dependency. Our model results suggest that subsidizing European module production would both support the growth of the local PV industry and reduce local supply chain emissions. Subsidies for producers are conducive to increasing the self-sufficiency in module production by shifting some costs from companies to the government. In Europe, supporting local solar PV manufacturing through subsidies or investment can improve competitiveness, create jobs, and increase self-reliance. Compared to trade barriers on China, subsidies can cut industry expenses by 23.6%, create jobs 27.5% more cost-efficiently, and provide a similar gain in self-sufficiency.

Performing a Monte Carlo analysis (Methods—Sensitivity analysis) shows that our results are robust against the uncertainty in cost data and expected demand. Global regional production shares remain similar throughout the sensitivity analysis, particularly in scenarios prioritizing global objectives. Demand levels directly influence the total manufacturing capacity. For example, a ±50% change in total global demand leads to corresponding changes of −50% to +67% in 2030 capacity (Supplementary Fig. 11). Meanwhile, changing cost assumptions amplifies the differences in national-level market shares between scenarios (Supplementary Fig. 12). For example, aiming to increase job production in Europe leads to greater variability in European production shares as compared to other scenarios (see ROE and Germany in Supplementary Fig. 12). Cost fluctuations in individual products do not significantly affect regional market shares; however, market shares are more sensitive to variations in downstream costs than upstream costs (Supplementary Figs. 13, 14). The total job creation potential also varies strongly depending on the regional demand data that drive the model. Accordingly, while our model relies on similar labor intensities (jobs/GW) as other studies, the total job

creation estimates may vary. For example, our results suggest that Europe could host over 125,000 full-time equivalent (FTE) jobs and 74 GW of module production by 2028, whereas industry reports[44] suggest roughly 60,000 FTE for 30 GW of module production. Differences in input data (e.g., the demand data that drives the model) and methodology lead to the gaps between our results and other reports; differences in output highlight the uncertainty surrounding future supply chain planning.

Efforts to boost self-sufficiency must consider what parts of the supply chain to try to localize. Subsidizing module production is often the most cost-effective and feasible focus, with capacity expansion for other parts of the value chain costing up to ten times more per kW of end product (Supplementary Table 1). In any case, funding for PV manufacturing must be justified against other public priorities and energy policy goals. For example, although the EU seeks greater self-sufficiency in PV manufacturing, it is also committed to being fully independent of Russian oil and gas[45]. Policymakers considering subsidies must also account for subsidy timing to best support their markets: governmental subsidies are most effective at the early exploratory stage rather than when local industry is mature[46], presenting different strategies for developed economies and emerging markets. From the perspective of industrial costs and job creation, our results align with the European Commission's recommendation that a "mixed strategy"—retaining necessary imports and diversifying suppliers—is essential for decarbonization, particularly when Europe is at a cost disadvantage[9,16,41].

International trade supports global renewable deployment goals in several ways. First, a global PV market helps regional capacity development and utilization by distributing global demand towards the least-cost suppliers and available manufacturing capacity. By extension, a global market also facilitates a fast transition by allowing regions immediate access to clean energy technologies even as they strive to build up their local production capacity. Second, global collaboration enables technology transfer, standardizes regulations, and maintains controllable costs (as modeled through assumed technology learning rates)[47]. These developments also support human capacity growth, as building up local manufacturing capacity requires not only new capacity investments but also technology and skill training. Finally, a global supply chain is more robust against shocks, e.g., suddenly imposed trade barriers, which could derail the global energy transition.

Achieving globally diversified supply chains requires both international cooperation and local efforts. One key requirement is common manufacturing and quality standards, such that components have larger global markets; stable and efficient communication channels between producers and purchasers would also help to build a resilient supply chain. Companies and governments must also provide the skills and technology training needed to build up a local workforce. Achieving a global transition requires all regions to increase the size of their workforces, but regional efforts may differ considerably. Our results suggest that European and American policymakers need to increase their workforces by more than 120% each year to meet supply chain targets, while Asian regions require annual job growth below 30% (see Supplementary Table 2). Overall, localizing and maintaining PV supply chains will depend not only on investment, but also on rapidly expanding the available workforce.

Global supply chains also feature strong environmental and social trade-offs. For instance, localizing production in developed economies cuts emissions but reduces global jobs and opportunities for developing economies to benefit from the low-carbon industry. Accounting for regional labor and environmental policies could further increase the global social and environmental trade-offs between local and international supply chains. For example, accounting for the EU's Carbon Border Adjustment Mechanism[25] would reduce the cost advantages of importing PV made in carbon-intensive economies and

further strengthen the argument for localizing the PV industry. However, a more self-reliant Europe also reduces job opportunities and low-carbon technology development in Southeast Asia. Nonetheless, the trade-offs offered by globalized supply chains would change over time. Global cooperation could help reduce regional differences in labor intensities, and global decarbonization will reduce regional differences in the carbon intensity of manufacturing. Engaging in global cooperation paradoxically also increases regions' competitive positions as they acquire infrastructure, technological expertise, and a skilled workforce.

As a cornerstone of the net-zero emissions energy system, installing solar PV requires a stable and reliable supply, and transparent assessments of costs, carbon emissions, and employment impacts. This research lays the groundwork for understanding the dynamics between producers and consumers and the consequences of price-led supply chain fluctuations on global climate objectives by providing a comprehensive mapping of the global PV supply chain's transformation and its associated impacts. However, our work has some limitations that ought to be addressed in future work. First, the model simplifies regional strategies to achieve a globally optimized solution, which enables us to explore cost-and job-optimal supply chain transitions under policy scenarios and assess trade-offs. However, it does not capture the full complexity of individual companies' actions or country-specific policies, and our results should not be seen as projections. Real-world development is strongly influenced by political will[1], infrastructure and capacity availability[1,9], and investment risk[48]. Future work could account for these factors by incorporating their consideration into the optimization process or by investigating likely development pathways[49,50]. Additionally, our research focuses on the regional level and uses yearly data, thereby not capturing within-region trade or sub-yearly trade fluctuations. Future studies with higher temporal and spatial resolution would deepen the understanding of renewable energy technology supply chains, better distribute the supply chain benefits, and identify supply chain links most vulnerable to political or natural disruptions. Finally, resolving data gaps in terms of product demand, trade flows, and quality would provide a fuller picture of the current supply chains and facilitate more nuanced analysis in terms of actors, space, and time (see Data sources). Solving these data gaps requires updating trade accounting standards to more precisely capture the international trade of low-carbon technology. Combined, this future work would help characterize current supply chains and help governments identify the supply chain strategies needed to achieve a fast, low-cost, and resilient low-carbon energy transition.

## Methods

### Modeling framework

A linear programming optimization model is developed for the solar PV global supply chain analysis. Unlike other possible modelling approaches[10,49], our optimization model explicitly seeks out solutions that maximize a specified goal. As such, the results help establish the boundaries on future possibilities rather than establish what is per se likely to occur.

Here, the global market is separated into 12 regions, including the top eight producers of solar PV (mainland China - CHN, Vietnam - VNM, United States - USA, Malaysia - MYS, Germany - DEU, Thailand - THA, Korea - KOR, and India - IND)[36], three aggregated regions (Rest of Europe - ROE, Rest of Asia - ROA, and Rest of World - ROW), and Switzerland (CHE) as a typical region that only possesses manufacturing capacity of one step in the supply chain (i.e., for one product). In the main text, Europe (EUR) indicates the region that includes Germany, Switzerland, and ROE. Each region is modeled as a node with PV demand, production capacity, and production costs. The supply chain itself considers the production of solar PV's five main components: polysilicon, ingots, wafers, cells, and modules. Producing each

component requires input from lower-value components; namely, producing modules requires cells, producing cells requires wafers, and so on (as shown in Fig. 1a and Supplementary Fig. 1).

Each of the 12 regions can fulfill their per-component demand by manufacturing the products themselves, using reserve stocks, or importing components from other regions. Similarly, the regions can export components to other regions. Each region can invest in local manufacturing by paying an upfront capital cost. This cost depends on the amount of new capacity built and the regional cost of expanding capacity. Production costs are based on the actual output and per-unit production cost, but decline with global cumulative PV manufacturing capacity, reflecting the technology's learning rate[7]. Inter-regional trade costs include transportation costs, communication costs, and costs to comply with foreign regulations[51]. These costs are modeled using a unified method that estimates trade costs and their effects on economic agents[43,51].

**Decision variables.** Decision variables in the model include capacity expansion, production levels, trade flows, and inventory management. For the product $p$ in region $i$ on year $T$, capacity investment ($\text{CapInv}_i^{p,T}$) determines how much new production capacity to build in each region and time period. This decision is informed by lead times, historical expansion trends, and projected demand. Capacity ($\text{Cap}_i^{p,T}$) reflects the cumulative effect of past investments, accounting for delays before new facilities come online. Production ($y_i^{p,T}$) is determined to meet demand while respecting capacity limits and job creation targets. Trade flows ($x_{i,j}^{p,T}$) allocate products across regions (from $i$ to $j$), balancing cost-effective shipping with demand fulfillment. Inventory ($\text{Stock}_i^{p,T}$) acts as a buffer, smoothing supply-demand gaps across time periods.

**Supply chain parameters.** The supply chain is calibrated using five types of parameters. Cost parameters include export costs ($\text{EC}_i^{p,T}$), import costs ($\text{IC}_i^{p,T}$), production costs ($\text{ProdCost}_i^{p,T}$), capacity expansion expenses ($\text{CapCost}_i^p$), and stock costs ($\text{StockCost}_i^{p,T}$). Subsidies ($\text{Subsidy}_i^{p,T}$) is enabled in some scenarios to reduce industrial producers' production costs. These cost parameters directly influence the affordability of each purchase and production decision. Employment factors ($\text{Job}_i^p$) tie production levels to job creation and are measured in full-time jobs per unit production of $p$. Technical constraints, like lead times ($Tl_i^p$) and historical expansion ceilings ($\text{CapLim}^p$), ensure solutions are actionable.

Material conversion factors capture demand relationships. Specifically, the conversion factors ($\text{Conversion}^{p1,p2}$) capture the ratio of material $p1$ needed to produce product $p2$ and are used for the material demand ($\text{Demand}_i^{p1,T}$) projections. For example, polysilicon demand can be calculated by multiplying the amount needed to produce one unit of ingot by the total ingot production. Emission factors for production ($\text{ProdEF}_i^p$) and trade ($\text{TradeEF}_{i,j}^p$), are applied to the production[4] and trade[52] of each component. We focus on one example of social and environmental outputs—jobs and carbon emissions—and their variation between locations. Other impacts of real-world production, such as labor standards and pollution control, are beyond the scope of this study.

**Objectives.** We consider two types of objectives, minimizing costs and maximizing jobs, which are considered in bi-objective optimizations (Eq. (1)). First, we optimize the global total costs (Eq. (2)), where the model is tasked with finding the optimal solution by varying trade flows ($x_{i,j}^{p,T}$) and production ($y_i^{p,T}$). The results from this optimization (Eq. (2)) provide the reference values for cumulative costs ($\text{Costs}_0$) and associated job creation ($\text{Jobs}_0$) used in Eq. (1). We then take the reference values for the bi-objective optimization (calculated by Eqs. (1–3)) with weights from 0 to 1 by a step of 0.1 each, which led to 59 unique possible supply chains. Here, normalized cost and job metrics

**Table 1 | Objectives and constraints for scenarios shown in the main text**

| Objective beside Minimize global costs (Eq. (2)) | Additional constraints for policy cases | Scenario names |
|---|---|---|
| Maximize global jobs (Eq. (3)) | None | Baseline |
| Maximize global jobs (Eq. (3)) | Trade barrier | European trade barrier |
| Maximize global jobs (Eq. (3)) | Subsidy | European module subsidies |
| Maximize European jobs (Eq. (3)) | None | MEJ |
| Maximize European jobs (Eq. (3)) | Trade barrier | MEJ + Trade barrier |
| Maximize European jobs (Eq. (3)) | Subsidy | MEJ + Subsidy |
| Maximize European jobs (Eq. (3)) | Self-sufficiency in 2030 | MEJ + Suff |
| Maximize European jobs (Eq. (3)) | Trade barrier + Self-sufficiency in 2030 | MEJ + Trade barrier + Suff |
| Maximize European jobs (Eq. (3)) | Subsidy + Self-sufficiency in 2030 | MEJ + Subsidy + Suff |

*Eq* equation, *MEJ* maximize European jobs, *Suff* self-sufficiency.

are weighed (with $w_a$ for cost and $w_b$ for jobs), allowing decision-makers to explore trade-offs. For instance, a higher $w_b$ would favor job-rich but potentially costlier production strategies. To reconcile these competing objectives, the model combines them into a single bi-objective function:

$$\min Z_c = w_a \frac{Z_a}{Costs_0} - w_b \frac{Z_b}{Jobs_0} \quad (1)$$

In the baseline case, the maximum job creation is found for all 12 regions. In the Maximize European Jobs (MEJ) policy scenarios, only the jobs created in Europe are considered within $Z_b$, i.e., $i \in$ Europeanregions.

The first objective, affordability ($Z_a$), aims to minimize total system costs (Eq. (2)). Costs include export and import expenses for inter-region trade ($EC_i^{p,T}$ and $IC_j^{p,T}$), production costs to industry, inventory holding costs, and capital expenditures for expanding manufacturing capacity.

$$
\begin{aligned}
\min Z_a = & \sum_T \sum_p \sum_i \sum_j \left( EC_i^{p,T} + IC_j^{p,T} \right) \cdot x_{i,j}^{p,T} \\
& + \sum_T \sum_p \sum_i ProdCost_i^{p,T} \cdot (1 - Subsidy_i^{p,T}) \cdot y_i^{p,T} \\
& + \sum_T \sum_p \sum_i StockCost_i^{p,T} \cdot Stock_i^{p,T} \\
& + \sum_T \sum_p \sum_i CapCost_i^{p} \cdot \left( Cap_i^{p,T} - Cap_i^{p,T-1} \right)
\end{aligned}
\quad (2)
$$

The second objective, job creation ($Z_b$), aims to maximize employment by prioritizing production activities that generate the most jobs per unit output (Eq. (3)). The number of full-time jobs created depends on production levels ($y_i^{p,T}$) and the region and product-specific employment factor ($Job_i^p$)[4].

$$\max Z_b = \sum_T \sum_p \sum_i Job_i^p \cdot y_i^{p,T} \quad (3)$$

Finally, we select all scenarios with the supply chains costs no more than 110% of the lowest-cost solution[53]. By considering supply chains that are nearly cost-optimal, our analysis reveals the potential trade-offs between economic costs and job creation, thereby facilitating policy discussions.

**Constraints.** The model enforces several critical constraints to ensure realistic and feasible solutions. Supply-demand balance is maintained through two key equations: one linking intermediate product demand to production via conversion factors ($Conversion^{p1,p2}$; Eq. (4)) and another ensuring final demand is met through local production ($y_j^{p,T}$),

imports ($x_{i,j}^{p,T}$), exports ($x_{j,k}^{p,T}$), and available stock ($Stock_j^{p,T}$; Eq. (5)).

$$Demand_i^{p1,T} = y_i^{p2,T} \cdot Conversion^{p1,p2} \quad (4)$$

$$Demand_j^{p,T} = y_j^{p,T} + \sum_i x_{i,j}^{p,T} - \sum_k x_{j,k}^{p,T} + Stock_j^{p,T-1} - Stock_j^{p,T} \quad (5)$$

Capacity expansion follows a lead time-delayed process: investments announced at time $T$ only become operational after the product-specific lead time ($Tl_i^p$; Eq. (6)). Expansions are capped at 95% of historical rates ($CapLim^p$; Eq. (7)) to reflect challenges of scaling up new technologies[54].

$$Cap_i^{p,T} = Cap_i^{p,T-1} + \sum_{t \le T} CapInv_i^{p,t-Tl_i^p} \quad (6)$$

$$CapInv_i^{p,T} \le Cap_i^{p,T-1} \cdot CapLim^p \quad (7)$$

Production and trade limits prevent infeasible outcomes. Production must be positive and cannot exceed available capacity (Eq. (8)) and regional exports cannot exceed its production (Eq. (9)).

$$y_i^{p,T} \le CapNew_i^{p,T} \quad (8)$$

$$\sum_j x_{i,j}^{p,T} \le y_i^{p,T} \quad (9)$$

## Scenarios

The analysis considers two main objectives and three types of industry support policies (Table 1), which we investigate in a set of scenarios. The main policy goals are minimizing system cost and maximizing job creation, which we consider in two main scenarios. The baseline scenario optimizes the global supply chain to minimize industry costs and maximize job creation. The second main scenario is the Maximize European Jobs scenario, where we shift the emphasis from global to European job creation while still aiming to minimize global costs as the secondary objective.

In addition to the two main scenarios, we consider three sets of policy cases. The first policy case is "limits on trade". We consider two trade policy cases: the first case is free trade (no trade restrictions), and the second case is Europe stops importing all the components directly from China. These trade scenarios respectively mimic the free trade promoted by the World Trade Organization and the trade defense driven by geopolitical tensions. The trade disruptions begin in 2024 and limit the trade of product $p$ between a given trade pair, region $i$ and region $j$ ($i \ne j$) on year $T$ ($T > 2024$) by constraining the trade flow to

zero, as per Eq. (10).

$$x_{ij}^{pT} = 0 \qquad (10)$$

The second policy case is "manufacturing subsidies". Specifically, we consider subsidies to decrease production costs in Europe and stimulate domestic module production. The subsidies are worth 20% of production costs, which is approximately the relative cost variance between Europe-made and imported PV from literature[4,15]. We do not consider subsidies for the other, lower-value PV components (i.e., polysilicon, ingots, wafers, and cells). The competing subsidies are valid starting from 2024, which means that the government in region $i$ subsidizes the production of PV modules ($T$ >2023) to producers.

The final policy case is "European self-sufficiency targets for 2030". Here, self-sufficiency is defined as the share of local production relative to local demand (Eq. (11)), with a maximum value of 100% when local production exceeds local demand. This scenario mimics Europe's self-production targets proposed in European Net-Zero Industry Act[26]. We implement linear constraints that gradually increase from 0% to the target self-sufficiency between 2025 and 2030 (Eq. (12)). For example, if the European self-sufficiency target is 40%, the constraints will increase incrementally by 8% each year: 0% in 2025, 8% in 2026, 16% in 2027, 24% in 2028, 32% in 2029, and 40% in 2030. This policy case is modeled for scenarios with maximum job creation in Europe as an objective, since these two goals are linked in actual policy documents[26]. The model results about self-sufficiency in Europe are shown as Supplementary Fig. 15.

$$\frac{y_i^{pT}}{Demand_i^{pT}} \geq Suff_i^{pT} \qquad (11)$$

$$ss_i^{pT} = \frac{T - 2025}{5} Suff_i^{p2030} \qquad (12)$$

### Sensitivity analysis

We conduct a Monte Carlo simulation to assess the impacts of uncertain demand and costs on manufacturing capacity, regional global production shares, and cumulative costs needed to achieve self-sufficiency in Europe. We assume that the demand for PV modules across 12 regions varies independently between 50 and 150% of the baseline data, following a uniform distribution. Similarly, cost data, including capacity expansion, production, trade, and stock costs, vary uniformly from 50 to 150% of the model's baseline assumptions. After 100 random samples of uncertain demand and costs, we executed the bi-objective optimization model with 51 unique weights (down from 59 after rounding and redundancy checks, ensuring efficient but comprehensive coverage of the cost-employment trade-off space) across the four main scenarios—baseline, maximize European jobs, subsidies in Europe, and trade barrier between Europe and China. Then, we compare all scenarios with the supply chain costs no more than 110% of the lowest-cost solution. The results of sensitivity in regions' market shares and manufacturing capacity are shown in Supplemental Information (Supplementary Figs. 11–13).

### Data sources

The analysis relies on a combination of open privately held data relating to current PV production patterns and forecast trends. We next present the data sources in the order in which they were integrated within the analysis.

First, existing demand and production capacity by country are collected from the BloombergNEF database[36], while future demand estimates correspond to the BloombergNEF high-demand net-zero scenario. The model's capacity expansion constraint in the model is the 95-percentile of the historical (2008–2020) national ratio between annual announced capacity and annual commissioned capacity based on this database. This information is privately held but can be accessed through the purchase of a user license.

Next, capacity expansion costs by region are collected from research by the National Renewable Energy Laboratory[42] and The European House[55]. PV production costs are collected from reports from the Department of Energy, Solar Energy Technologies Office of the United States[42,56,57], the International Energy Agency (IEA)[4], and the International Renewable Energy Agency's (IRENA)[58]. The production costs data covered major producers, including China, the United States, the Association of Southeast Asian Nations (ASEAN), India, Korea, and Europe. Future production costs are estimated assuming a decline with cumulative capacity and learning rates collected from literature[7] (details in Supplementary Data 1).

Trade cost indices by country, year, and sector, including export cost index (ECI) and import cost index (ICI), are collected from WTO research[43,51]. The trade cost index includes transportation costs, trade policy barriers, costs to comply with foreign regulations, communication costs, transaction costs, or information costs[43]. The COVID pandemic increased global trade costs in multiple ways: transport and travel costs have witnessed an increase due to the global logistics crisis, including port congestion, increasing shipping times, and the scarcity of containers, increased border controls, as well as the insufficient resilience of trade policies[59,60]. Therefore, we assume that the import and export cost indices in 2022 increased relative to 2018 levels, following existing literature[59].

Conversion factors between segments in PV supply chain, stocks of modules, lead time for manufacturing investment by region and product, and job creation of the manufacturing by product are collected from the Special Report for Solar PV Global Supply Chain from IEA[4]. These values are based on production efficiencies from the year 2020–2021 (details in Supplementary Data 1).

Finally, we estimate the manufacturing impact by considering carbon dioxide emission factors collected from the IEA Special Report for Solar PV Global Supply Chain[4]. Emission factors of trade by sector are calculated based on the carbon emissions data from CEADs datasets[61], IEA greenhouse gas emissions data[62], and the EMERGING global multi-regional input-output model[63,64].

Our analysis relies on single, secondary sources for demand, production, cost, and conversion factors. Our main sources include BloombergNEF, IEA reports, and WTO, which provide relatively reliable and broad coverage of global regions in the PV supply chain. Relying on secondary sources always risks introducing bias linked to the market interests of the data providers and a focus on dominant technologies and regions. However, relying on secondary sources is a necessary compromise given the limited availability of primary data. While our main sources are globally trusted institutions, their data and projections may differ in definition and scope from alternative sources, which can contribute to discrepancies between our model results and those featured in other reports. Mitigating these risks would require a comparative analysis between all input data, an activity that remains outside the scope of the present work.

### Data availability

The data used to develop the model in this study has been made available in Supplementary Data 1 as an electronic spreadsheet. The formatted data used to run the model, perform sensitivity analysis, and generate the figures in the main text is available on Zenodo[65] (https://doi.org/10.5281/zenodo.14260389).

### Code availability

R code to develop the model, analyze sensitivity, and make figures in the main text is available at Zenodo[65] (https://doi.org/10.5281/zenodo.14260389).

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

## Acknowledgements
We want to thank Paolo Gabrielli and the Reliability and Risk Engineering lab for discussions on this topic. C.C. and G.S. are part of the SPEED2-ZERO, a Joint Initiative co-financed by the ETH Board.

## Author contributions
C.C. and G.S. designed the research. C.C. collected data, developed the model, and conducted the analyses with input from K.E.L. and G.S.. C.C. wrote the manuscript with input from K.E.L. and G.S.

## Funding

## Competing interests
The authors declare no competing interests.
