## [Transparent Peer Review file · Nature Communications]

Policy-driven transformation of global solar PV supply chains and resulting impacts

Corresponding Author: Professor Giovanni Sansavini

Version 0:

Reviewer comments:

Reviewer #1

(Remarks to the Author)

The manuscript presents three core conclusions. First, future global photovoltaic manufacturing will continue to rely heavily on China. Second, there is the potential to achieve simultaneous improvements in employment and PV self-sufficiency in some regions. Third, the paper calls for a more open international trade market and trade policy, which can reduce costs and ensure security as well as environmental benefits.

These findings offer a strong rationale for the EU and regions considering trade barriers to support a win-win photovoltaic manufacturing chain, thereby ensuring steady growth in new energy installation capacity along with economic, emission reduction, and employment benefits. The methodology is straightforward, the conclusions are robust, and the perspectives are innovative. With minor revisions as outlined below, the paper would be suitable for publication.

1. The authors stated in the last paragraph of the Introduction “We develop scenarios ... using a least-cost optimization model.” While in the Results and Methods sections, the model is described as bi-objective optimization, considering both costs and job creation. Please check and ensure consistency in how the methods are described.
2. Efficiency is an important aspect when analyzing capacity and production (after Figure 2). The authors note that “This uneven efficiency reveals that regional imbalances persist as regions prioritize low industry costs and local supply, resulting in underutilized capacity and wastage.” It would be valuable to further elaborate on this point in the discussion.
3. Regarding uncertainty, the authors wrote: “In comparison to cost uncertainty, demand uncertainty has the greatest effect on how much capacity is installed rather than the scenario-to-scenario variability of regional capacity shares.” Do they mean manufacturing capacity? If so, using a different term instead of “installed” could help avoid misunderstandings.
4. The manuscript has a tendency to describe the study as global supply chain considering the trade barriers; however, the scenario involving local trade barriers appears to consider only the EU. However, for example, the USA’s IRA requires a localization rate of over 50% for components, and as the USA represents a huge market for PV demand, it is important to discuss what impact further enhanced trade protection in the USA might have on global supply chain. Or Authors could polish the title of the manuscript to make it more precise
5. In the description above Figure 3, the text states: “Finally, government subsidies in Europe, amounting to 7.1% of the baseline industry costs, help share the financial burden on European industry (0.8% globally) but lower the global average self-sufficiency by 0.3%.” Could the authors expand the explanation and discussion regarding why the global average self-sufficiency is lowered?
6. Regarding Fig. S3 Manufacturing capacity and production in 2022 and by scenario in 2030, by product and region, the figure discusses manufacturing capacity and production (efficiency levels) across various processes and scenarios in different countries. My question is, taking China as an example, why does the baseline scenario for 2030 show such significant overcapacity? What is the underlying mechanism and reason? In other words, why have companies not adjusted their production capacity expansion in a timely manner?

Some minor suggestions for the figures:

7. In Figure 1c, there is an issue with the legend. The text states, “For example, module production in China and the US can become 1.5 times and 3.8 times the level in 2022, respectively,” but this multiplication factor is not evident from the figure. The differences in color intensity should be enhanced to more clearly reflect differences, especially between the ranges of 0–100% and 100–200%.
8. For Figure 2, although the chord diagram effectively displays the import-export status, it could be improved for readers unfamiliar with this type of diagram. The arrows indicating the flow direction should be made clearer. For instance, the purple

flow from China to China is so large that the arrow direction is not obvious. It might be helpful to add a directional arrow line in the center of each flow band to clearly indicate the movement from the origin to the destination.

9. In Figures 3A–D, it is unclear whether the ranges represent the impacts of various indicators for each country, with the solid points representing the medians. Additionally, there is a minor error in the caption of Figure 3, where two consecutive commas appear on the fourth line.

10. The authors have provided helpful supplementary figures to support the main text. However, some scenario names in the supplementary figures do not match those in the main text. Please review and align them for better readability across all documents.

(Remarks on code availability)

Reviewer #2

(Remarks to the Author)

The study examines the future transformation of the global photovoltaic (PV) solar supply chain from 2021 to 2030, highlighting China's continued dominance, particularly in lower-value PV components. It emphasizes that regional supply chain policies can boost job creation and self-sufficiency, but supply chain diversification may increase costs, especially in high-labor-cost regions requiring new production facilities. Open trade policies are deemed essential for minimizing costs while balancing security and environmental concerns. Using an optimization-based model, the study evaluates different policy-driven scenarios and their financial, environmental, and job creation implications, demonstrating how regional strategies influence global market dynamics and climate goals. The methodology is robust, with transparent decision variables and constraints. While some limitations exist, the study provides critical findings for policymakers, industry leaders, and researchers interested in solar energy supply chains and the broader energy transition in a changing geopolitical and trade landscape. Despite its valuable insights, the study simplifies regional strategies to achieve a globally optimized solution, potentially overlooking country-specific policies.

I noticed some minor shortcomings in the text:

1. The introduction requires rewriting and grammatical improvements. There are punctuation errors in the second paragraph that make some sentences unclear.
2. The third paragraph is confusing and difficult to follow. Specifically, some standalone sentences are not well connected to the overall flow and main argument. I recommend a complete rewrite with a clearer structure—starting with the central idea (China's dominance is perceived as a barrier, prompting the EU and the USA to develop counterstrategies), followed by supporting arguments (enhancing competitiveness, increasing diversification, and strengthening supply chain security).
3. In the paragraph beginning with "Recent work has made significant progress in identifying existing supply chain patterns...", I suggest including specific examples of these findings and avoiding the word "significant" without supporting evidence. Likewise, when discussing remaining gaps, specify some of them and clarify how your research contributes to the existing body of literature.
4. I recommend improving the introduction's connection to the problem statement. While the analysis and results sections are clear and concise, the introduction feels somewhat disconnected. Given the topic's high relevance in a shifting economic and geopolitical landscape, the introduction should be more engaging to better capture the reader's attention.

(Remarks on code availability)

Although the record is publicly available, I had no access to the files as this are restricted.

Reviewer #3

(Remarks to the Author)

The manuscript presents an in-depth analysis of the global solar PV supply chain and explores potential transformations driven by economic and policy considerations. The topic is highly relevant to the ongoing energy transition and supply chain security concerns. The methodology is well-structured, leveraging optimization modeling to analyze different scenarios. However, there are several areas where clarity, rigor, and completeness could be improved.

1. The manuscript presents various scenarios, but some assumptions lack clear justification. For instance, the selection of subsidy levels (20% for Europe) and self-sufficiency targets (e.g., 40%) should be more thoroughly justified with references to policy documents or industry trends.
2. The rationale behind capacity expansion constraints (historical maximum annual percent increment) needs to be explained more explicitly. Have these constraints been validated against industry growth trends?
3. The manuscript relies heavily on BloombergNEF and IEA data, but more discussion on data limitations and potential biases is necessary.
4. How were cost projections validated? Are they consistent with recent PV industry reports beyond the cited sources?

5. A comparison with past projections vs. actual outcomes would strengthen the reliability of the model.
6. The study considers five main scenarios, but the European-focused scenarios should incorporate more nuanced policy variations. For example, how would different levels of trade restrictions (partial vs. full bans) alter the outcomes?
7. The impact of geopolitical events (e.g., U.S.-China trade tensions) on supply chain resilience is not fully explored. Would alternative supply chains emerge in Southeast Asia or India under stricter trade restrictions?
8. The manuscript should better address the limitations of the optimization approach. While it identifies cost-optimal solutions, real-world constraints such as political will, infrastructure availability, and investment risk should be discussed.
9. The sensitivity analysis is helpful but could be expanded. For example, how do variations in raw material costs (e.g., polysilicon price fluctuations) impact the scenario outcomes?
10. The study mentions carbon emissions and job creation, but a deeper discussion on social and environmental trade-offs is needed.
11. Does the model account for environmental regulations affecting manufacturing locations (e.g., stricter labor and pollution controls in Europe vs. China)?
12. How do job creation estimates compare with actual labor market conditions in different regions?
13. Figure 2 and Figure 3 contain valuable insights but require clearer labeling and captions.
14. Some sentences, especially in the introduction and discussion, are overly long and complex. Consider simplifying key messages for better readability.

(Remarks on code availability)

Version 1:

Reviewer comments:

Reviewer #1

(Remarks to the Author)

The author has made substantial revisions and has thoroughly addressed all of my concerns. I believe the manuscript is now suitable for publication.

(Remarks on code availability)

Reviewer #2

(Remarks to the Author)

After reviewing the revised manuscript and the author's responses to my earlier comments, I am satisfied that all concerns have been fully addressed. The authors have clearly incorporated the suggested improvements, increasing the clarity and rigor of the manuscript. The introduction alligns with the methodology and analysis and provides clear flow between the sections and the overall view of the topic under analysis.

(Remarks on code availability)

Code is clear and understandable.

It follows the documentation guidelines and provides clearly every step produced for the analysis.

Reviewer #3

(Remarks to the Author)

The authors appear to have satisfactorily addressed my comments and suggestions in their revised manuscript.

(Remarks on code availability)

Response to reviewers

We would like to thank the reviewers for their time and thoughtful, constructive comments. We have addressed each comment point-by-point and would like to highlight the following changes:

- We have rewritten the introduction to be more engaging and more directly linked to our research questions
- We have added new scenarios to better comment on how increasing trade restrictions could further alter global supply chains
- We have reorganized the discussion and methods sections to improve clarity and readability.
- We have extended the sensitivity analysis to show how cost changes in individual products influence market shares.

All changes are highlighted in a marked-up version of the manuscript, with revised or added text shown in blue and deleted text shown with a ~~red strikethrough~~.

Thank you again for your time and constructive feedback. We believe that the quality of the work has benefited from your insight, especially with regards to clarity and positioning our work in relevance to real-world events. We hope that you approve of the modifications.

Kind regards,

The Authors

REVIEWER COMMENTS

Reviewer #1 (Remarks to the Author):

R1.0. The manuscript presents three core conclusions. First, future global photovoltaic manufacturing will continue to rely heavily on China. Second, there is the potential to achieve simultaneous improvements in employment and PV self-sufficiency in some regions. Third, the paper calls for a more open international trade market and trade policy, which can reduce costs and ensure security as well as environmental benefits. These findings offer a strong rationale for the EU and regions considering trade barriers to support a win–win photovoltaic manufacturing chain, thereby ensuring steady growth in new energy installation capacity along with economic, emission reduction, and employment benefits. The methodology is straightforward, the conclusions are robust, and the perspectives are innovative. With minor revisions as outlined below, the paper would be suitable for publication.

Thank you for your time and comments!

R1.1. The authors stated in the last paragraph of the Introduction “We develop scenarios ... using a least-cost optimization model.” While in the Results and Methods sections, the model is described as bi-objective optimization, considering both costs and job creation. Please check and ensure consistency in how the methods are described.

Thank you for your comment. We have revised the description of the model in Introduction using consistent words with the latter description.

We test different European policy scenarios using a bi-objective optimization model considering costs and job creation and analyze the corresponding impacts over a period of 2021 to 2030 (Methods – Modelling framework).

R1.2. Efficiency is an important aspect when analyzing capacity and production (after Figure 2). The authors note that “This uneven efficiency reveals that regional imbalances persist as regions prioritize low industry costs and local supply, resulting in underutilized capacity and wastage.” It would be valuable to further elaborate on this point in the discussion.

Thank you for your comment and interest. We have further elaborated on this point in Discussion:

International trade supports global renewable deployment goals in several ways. First, a global PV market helps regional capacity development and utilization by distributing global demand towards least-cost suppliers and available manufacturing capacity. By extension, a global market also facilitates a fast transition by allowing regions immediate access to clean energy technologies even as they strive to build up their local production capacity.

R1.3. Regarding uncertainty, the authors wrote: “In comparison to cost uncertainty, demand uncertainty has the greatest effect on how much capacity is installed rather than the scenario-to-scenario variability of regional capacity shares.” Do they mean manufacturing capacity? If so, using a different term instead of “installed” could help avoid misunderstandings.

Thank you for your comment. We revised the wording here to avoid a misunderstanding:

Demand levels directly influence the total manufacturing capacity.

R1.4. The manuscript has a tendency to describe the study as global supply chain considering the trade barriers; however, the scenario involving local trade barriers appears to consider only the EU. However, for example, the USA’s IRA requires a localization rate of over 50% for components, and as the USA represents a huge market for PV demand, it is important to discuss what impact further enhanced trade protection in the USA might have on global supply chain. Or Authors could polish the title of the manuscript to make it more precise.

Thank you for your comment. Although our research focus is on the impacts of European policy action on global supply chains, we agree that the trade between the United States of America and China is influential to global supply chain of PV. To better contextualize our work and recognize the potentially complementary role of American trade barriers, we now test additional scenarios where both the USA and Europe ban imports from China. We have amended the findings and discussion correspondingly:

Similar effects can be expected if both the USA and Europe implement trade barriers against China – the unmet demand turns to India, Malaysia, Vietnam, and ROA (Supplementary Fig. 3). Banning products along the entire supply chain has greater impacts in rerouting supply chains than banning individual products (Supplementary Fig. 3). Moreover, banning the trade of downstream products simply shifts the dependence of upstream products on other regions, i.e., banning module imports leads to more imports of cells (Supplementary Fig. 3).

Please also see our response to R3.7.

R1.5. In the description above Figure 3, the text states: “Finally, government subsidies in Europe, amounting to 7.1% of the baseline industry costs, help share the financial burden on European industry (0.8% globally) but lower the global average self-sufficiency by 0.3%.” Could the authors expand the explanation and discussion regarding why the global average self-sufficiency is lowered?

Thank you for your comment. The global average self-sufficiency declines because the build-up of European industry reduces the continent’s dependence on other regions and reduces the need for other regions to increase domestic production,

This effect occurs because localizing European production reduces dependence on other regions and even reduces the need for other regions to build up their manufacturing capacity. In turn, this also reduces the ability of other regions to meet their local demand with local supply. These results collectively indicate the power of regional (e.g., European) policies on global supply chains and their resulting global socioeconomic and environmental impacts.

R1.6. Regarding Fig. S3 Manufacturing capacity and production in 2022 and by scenario in 2030, by product and region, the figure discusses manufacturing capacity and production (efficiency levels) across various processes and scenarios in different countries. My question is, taking China as an example, why does the baseline scenario for 2030 show such significant overcapacity? What is the underlying mechanism and reason? In other words, why have companies not adjusted their production capacity expansion in a timely manner?

Thank you for raising this point. We checked the model results and found that the previous Fig. S3 contained a mistake and showed the *average* capacity and production of all weights of bi-objective costs and jobs optimization instead of the near-optimal weights that are consistent with the main text results. As shown in the updated (correct) figure below, China has overcapacity because China supplies for other regions' demand when their capacity is insufficient. The construction of new capacity in other regions takes more time than in China (see lead time in Supplementary Data).

Supplementary Figure 1 Manufacturing capacity and production in 2022 and by scenario in 2030, by product and region. China has a surplus capacity because it supplies other regions when they cannot meet their own demand because building new capacity in those regions takes longer than China.

Some minor suggestions for the figures:

R1.7. In Figure 1c, there is an issue with the legend. The text states, “For example, module production in China and the US can become 1.5 times and 3.8 times the level in 2022, respectively,” but this multiplication factor is not evident from the figure. The differences in color intensity should be enhanced to more clearly reflect differences, especially between the ranges of 0–100% and 100–200%.

Thank you for your comment. We agree that enhancing the color intensity would more clearly reflect differences. Figure 1c is now revised with more scales of relative increases:

Fig. 1 Schematics of the model and definitions of key elements, basic processes, and regions. Panel a shows the model framework with 12 regions participating in five components' supply, based on the demand and manufacturing capacity data in 2021. Point size shows the relative module demand and production capacity. Panel b shows the options for meeting increasing demand within a region, with time series from year t to several years' lead time (L) beyond year t . The process is shown for a sample region. Panel c shows the relative increase in module production from 2022 to 2030 for the modelled baseline scenario.

R1.8. For Figure 2, although the chord diagram effectively displays the import-export status, it could be improved for readers unfamiliar with this type of diagram. The arrows indicating the flow direction should be made clearer. For instance, the purple flow from China to China is so large that the arrow direction is not obvious. It might be helpful to add a directional arrow line in the center of each flow band to clearly indicate the movement from the origin to the destination.

Thank you for your feedback. We appreciate that additional support could help readers unfamiliar with chord diagrams understand the figure. To do so, we have added new annotation: we have added a schematic arrow and have noted the production and demand in the outer ring and adjusted the arrow borders to more clearly show the arrow directions. We have also extended the figure caption.

Fig. 2 Overview of global PV supply chains as shown by region-to-region trade flows for individual PV products. Outer rings compound production and demand for the ten indicated regions. Arrows indicate regional trade flows and show the share of supply and demand for each outer ring; for example, the black arrow in (a) emphasizes that the majority of China's polysilicon supply is used to meet the domestic national demand and the remaining demand is mainly supplied from Europe and Malaysia. (a) shows 2022 trade flows and (b) shows the 2030 trade flows according to the tested scenarios. Trade flows for individual products are shown left-to-right following the PV value chain, with modules being the products sold to end-consumers.

R1.9. In Figures 3A–D, it is unclear whether the ranges represent the impacts of various indicators for each country, with the solid points representing the medians. Additionally, there is a minor error in the caption of Figure 3, where two consecutive commas appear on the fourth line.

Thank you for your comment. In Figures 3a-d, the ranges represent the global impacts of the four transition scenarios instead of the impacts of indicators for each country. For example, the global costs of PV supply chain under scenario “maximizing European job creation” is 3.68 trillion USD in median, instead of meaning maximizing European job creation costs 3.68 trillion USD more than maximizing global job creation (baseline). We also revised the figure labelling, legend, and caption to provide greater reader clarity.

Fig. 3 The cost and impacts of European policy actions on PV supply chains. Panels (a-d) show the scenario-by-scenario results on global (a) cumulative costs, (b) cumulative emissions, (c) cumulative job creation, and (d) average self-sufficiency across years. Panels (e-g) show the cumulative cost-jobs trade-offs according to policy actions aiming to (e) maximize European jobs (MEJ), (f) maximize European jobs and provide a production subsidy, and (g) maximize European jobs and impose a trade barrier on all Chinese PV products. The point shading in panels (e-g) corresponds to different European self-sufficiency targets, which are modelled in the optimization model as a constraint.

R1.10. The authors have provided helpful supplementary figures to support the main text. However, some scenario names in the supplementary figures do not match those in the main text. Please review and align them for better readability across all documents.

Thank you for your comment. We checked and aligned the scenario names in the supplementary figures.

Reviewer #2 (Remarks to the Author):

R2.0. The study examines the future transformation of the global photovoltaic (PV) solar supply chain from 2021 to 2030, highlighting China's continued dominance, particularly in lower-value PV components. It emphasizes that regional supply chain policies can boost job creation and self-sufficiency, but supply chain diversification may increase costs, especially in high-labor-cost regions requiring new production facilities. Open trade policies are deemed essential for minimizing costs while balancing security and environmental concerns. Using an optimization-based model, the study evaluates different policy-driven scenarios and their financial, environmental, and job creation implications, demonstrating how regional strategies influence global market dynamics and climate goals. The methodology is robust, with transparent decision variables and constraints. While some limitations exist, the study provides critical findings for policymakers, industry leaders, and researchers interested in solar energy supply chains and the broader energy transition in a changing geopolitical and trade landscape. Despite its valuable insights, the study simplifies regional strategies to achieve a globally optimized solution, potentially overlooking country-specific policies.

Thank you for your time and constructive feedback!

I noticed some minor shortcomings in the text:

R2.1. The introduction requires rewriting and grammatical improvements. There are punctuation errors in the second paragraph that make some sentences unclear.

Thank you for your feedback – we apologize for the errors. To address this comment and the feedback provided in R2.2-R2.4, we have performed a major rewrite of the introduction. In addition, we have reviewed the entire section with Grammarly to mitigate any grammar errors.

R2.2. The third paragraph is confusing and difficult to follow. Specifically, some standalone sentences are not well connected to the overall flow and main argument. I recommend a complete rewrite with a clearer structure—starting with the central idea (China's dominance is perceived as a barrier, prompting the EU and the USA to develop counterstrategies), followed by supporting arguments (enhancing competitiveness, increasing diversification, and strengthening supply chain security).

Thank you for your comment. In line with your other comments, we have performed a major revision on the introduction. We have restructured the introduction to first introduce China's dominance, then explain why such dominance is viewed as problematic. We next explain strategies and policies for increasing self-sufficiency and present examples from the USA and the EU. We close the introduction by explaining the literature gaps (see also R2.3) and how our work works to address those gaps.

R2.3. In the paragraph beginning with “Recent work has made significant progress in identifying existing supply chain patterns...”, I suggest including specific examples of these findings and avoiding the word “significant” without supporting evidence. Likewise, when discussing remaining gaps, specify some of them and clarify how your research contributes to the existing body of literature.

Thank you for your feedback. We have revised how we introduce prior literature to be more precise and reference specific examples. Namely, we write,

Despite the extensive and varied policy support, it is unclear whether European policymakers will be able to achieve their ambitions for localizing PV supply chains. Building up low-carbon manufacturing is a complex task, requiring a supportive regulatory environment¹² and access to skilled labor³⁹. Supplier preferences⁴⁰ and cost differences⁴¹ also influence industry’s willingness to alter supply chain patterns. Singular events and policies can disrupt global supply PV chains, as did the global COVID pandemic¹⁶ and American trade policies targeting Chinese production⁴². Recent work suggests that relocating PV manufacturing outside of China could help reduce supply chain emissions⁴³ but that systemic financial support is required to overcome financial barriers²⁸. As such, the full scope of the opportunities, trade-offs, and impacts of European policy action are yet unclear.

We also add a new paragraph explicitly stating how our research contributes to the existing body of literature.

Our research contributes to the literature in three ways. First, we demonstrate the impact of regional policy action on future global supply chain networks, with and without coordinated action of partner regions. By doing so, we overcome the limitations of past work that take only a retrospective perspective^{2,3} or focuses only on local policy impacts without considering knock-on supply chain effects^{4,5}. Second, our results reinforce existing literature in suggesting that open trade policy is key to minimizing costs and creating jobs, even when considering global and local security and environmental impacts. Finally, we provide an in-depth analysis of the specific trade-offs facing European policymakers, thereby providing region-specific decision-making input that has hereto been lacking.

R2.4. I recommend improving the introduction’s connection to the problem statement. While the analysis and results sections are clear and concise, the introduction feels somewhat disconnected. Given the topic’s high relevance in a shifting economic and geopolitical landscape, the introduction should be more engaging to better capture the reader’s attention.

Thank you for your constructive comments pertaining to our introduction. In conjunction with feedback in R2.1-R2.3, we have performed a major rewrite of the introduction to make the introduction more engaging and improve upon the section’s clarity and conciseness.

We specifically address comments R2.1-R2.3 in the following ways:

- R2.1: We have edited the section with the help of Grammarly software.
- R2.2: We have adopted your suggested narrative with respect to framing our work, starting with the central idea that China’s dominance is perceived as a barrier, prompting the EU and the USA to develop counterstrategies.

- R2.3: We have modified our discussion of existing literature to include specific examples of recent work and be more precise in presenting research gaps for building up a European supply chain (page 4, lines 127-139 in the marked manuscript).
- R2.3: We have added a new paragraph at the end of the introduction specifically stating our work's contributions towards closing existing research gaps (page 5, lines 155-164 in the marked manuscript).

Reviewer #2 (Remarks on code availability):

Although the record is publicly available, I had no access to the files as this are restricted.

Thank you for raising this point. We have reviewed the upload and confirmed that the files are now available. We apologize for this error.

Reviewer #3 (Remarks to the Author):

R3.0 The manuscript presents an in-depth analysis of the global solar PV supply chain and explores potential transformations driven by economic and policy considerations. The topic is highly relevant to the ongoing energy transition and supply chain security concerns. The methodology is well-structured, leveraging optimization modeling to analyze different scenarios. However, there are several areas where clarity, rigor, and completeness could be improved.

Thank you for your comment. We appreciate your suggestions on improving clarity, considering scenarios with more diverse trade policies, and comparing the results with other reports. We made substantial changes to the Introduction, Discussion and Methods and updated corresponding analysis in Results with the reviewers' comments.

R3.1. The manuscript presents various scenarios, but some assumptions lack clear justification. For instance, the selection of subsidy levels (20% for Europe) and self-sufficiency targets (e.g., 40%) should be more thoroughly justified with references to policy documents or industry trends.

Thank you for your comment. We have revised the text to more thoroughly justify the assumptions in Methods and Introduction sections.

In Methods section, we explain the selection of subsidy levels (20% for Europe) with reference to existing regional differences in production costs:

The second policy case is "manufacturing subsidies". Specifically, we consider subsidies to decrease production costs in Europe and stimulate domestic module production. The subsidies are worth 20% of production costs, which is approximately the relative cost variance between Europe-made and imported PV from literature^{3,6}. We do not consider subsidies for the other, lower-value PV components (i.e., polysilicon, ingots, wafers, and cells).

In Introduction section we explained the selection of self-sufficiency targets (40%), which is based on the European Net Zero Industry Act:

The EU has goals to reach 30 GW of operational solar PV manufacturing and 40% self-production of net-zero technologies, including solar PV, by 2030⁷.

We also reference the goal in the Methods section:

Here, self-sufficiency is defined as the share of local production relative to local demand (Eq. 11), with a maximum value of 100% when local production exceeds local demand. This scenario mimics Europe's self-production targets proposed in European Net Zero Industry Act⁷.

R3.2. The rationale behind capacity expansion constraints (historical maximum annual percent increment) needs to be explained more explicitly. Have these constraints been validated against industry growth trends?

Thank you for your feedback. We have modified the text in two places to explain the rationale behind the capacity expansion trends more explicitly.

First, we explain the need to introduce a capacity expansion constraint when we introduce the modelling framework, citing literature focused on technology diffusion:

Expansions are capped at 95% of historical rates ($CapLim^p$; Eq. 7) to reflect challenges of scaling up new technologies⁸.

Second, we reference how the expansion constraint is validated against industry growth trends in the “Data sources” section:

The model's capacity expansion constraint in the model is the 95-percentile of the historical (2008-2020) national ratio between annual announced capacity and annual commissioned capacity based on this database.

R3.3. The manuscript relies heavily on BloombergNEF and IEA data, but more discussion on data limitations and potential biases is necessary.

Thank you for your comment. Based on your feedback, we extend the discussion on our data sources, their limitations, and the potential impact on results. We make these changes in the Discussion and Data Sources sections:

In Discussion, we write:

Additionally, our research focuses on the regional level and uses yearly data, thereby not capturing within-region trade or sub-yearly trade fluctuations. Future studies with higher temporal and spatial resolution would deepen the understanding of renewable energy technology supply chains, better distribute the supply chain benefits, and identify supply chain links most vulnerable to political or natural disruptions. Finally, resolving data gaps in terms of product demand, trade flows, and quality would provide a fuller picture of the current supply chains and facilitate more nuanced analysis in terms of actors, space, and time (see Data sources). Solving these data gaps requires updating trade accounting standards to more precisely capture international trade of low-carbon technology. Combined, this future work would help characterize current supply chains and help governments identify the supply chain strategies needed to achieve a fast, low-cost, and resilient low-carbon energy transition.

In Data sources, we write:

Our analysis relies on single, secondary sources for demand, production, cost, and conversion factors. Our main sources include BloombergNEF, IEA reports, and WTO, which provide relatively reliable and broad coverage of global regions in the PV supply chain. Relying on secondary sources always risks introducing bias linked to the market interests of the data providers and a focus on dominant technologies and regions. However, relying on secondary sources is a necessary compromise given the limited availability of primary data. While our main sources are globally trusted institutions, their data and projections may differ in definition and scope from alternative sources, which can contribute to discrepancies between our model results and those featured in other reports. Mitigating these risks would require a comparative analysis between all input data, an activity that remains outside the scope of the present work.

R3.4. How were cost projections validated? Are they consistent with recent PV industry reports beyond the cited sources?

Thank you for your comment. As an additional clarification, we would like to highlight our optimization model aims to showcase potential future scenarios but does not aim to forecast the supply transition. Nonetheless, we agree with your point that comparison is valuable to the reader to help contextualize our results. We identified few reports discussing the total investment requirements for the whole future PV supply chain, but found some reports estimating the capacity investment needed for a European supply chain. In Results - Transformation requirements and costs, we added the comparison:

We estimate that capital costs for European supply chain development to be in the range of 50-120 billion USD. Our cost estimates for the baseline scenario align with those suggested by the European Solar Industry Alliance⁴⁰ in terms of the financing required to increase downstream wafer, cell, and module capacity. However, our results developing a fully European PV value chain require far greater financing than currently called for in other proposals³⁹: developing polysilicon and ingot manufacturing requires an additional 20-52 billion USD.

R3.5. A comparison with past projections vs. actual outcomes would strengthen the reliability of the model.

Thank you for your comment. We agree with the value of comparing the model results to actual trade where possible. In the Results, we compare actual trade flows to those suggested by baseline model results:

The results of the optimization model suggest that the key product flows resemble the current supply chain, but there are still opportunities for the supply chain transition to be more cost effective and job-supporting (Supplementary Fig. 2).

Supplementary Figure 2 also includes an extended caption where we compare the statistical and model-based results.

Supplementary Figure 2 Global PV supply chain in 2022 based on statistics^{9,12,13} and model baseline results. The circles show the total global trade flow of the PV components, with the size of each arrow indicating the flow from exporters to importers and the colors representing individual regions. The model results are similar to the current supply chain in terms of major suppliers, but there is still space for the current supply chain to transition to the one with global economic costs and job creation optimized. The model simplifies regional strategies to achieve a globally optimized solution, which enables us to explore cost-and-job-optimal supply chain transitions under policy scenarios and assess trade-offs. However, it does not capture the full complexity of individual companies' actions or country-specific policies and results should not be seen as projections.

As additional clarification, we would like to highlight our optimization model aims to showcase potential future scenarios but does not aim to forecast the supply transition. We highlight this aim in-text in the Introduction, the Discussion, and the Methods:

In the Introduction, we write:

“We test different European policy scenarios using a bi-objective optimization model considering costs and job creation and analyze the corresponding impacts over a period of 2021 to 2030 (Methods – Modelling framework). Relying on an optimization model allows us to test “what-if” scenarios associated with strong policy action (see Methods).”

In the Discussion, we write:

However, our work has some limitations that ought to be addressed in future work. First, the model simplifies regional strategies to achieve a globally optimized solution, which enables us to explore cost-and-job-optimal supply chain transitions under policy scenarios and assess trade-offs. However, it does not capture the full complexity of individual companies' actions or country-specific policies and our results should not be seen as projections. Real-world development is strongly influenced by political will¹⁴, infrastructure and capacity availability^{11,14}, and investment risk¹⁵. Future work could account for these factors by incorporating their consideration into the optimization process or by investigating likely development pathways^{16,17}.

In the Methods, we write:

A linear programming optimization model is developed for the solar PV global supply chain analysis. Unlike other possible modelling approaches^{16,18}, our optimization model explicitly seeks out solutions that maximize a specified goal. As such, the results help establish the boundaries on future possibilities rather than establish what is per se likely to occur.

R3.6. The study considers five main scenarios, but the European-focused scenarios should incorporate more nuanced policy variations. For example, how would different levels of trade restrictions (partial vs. full bans) alter the outcomes?

Thank you for your suggestion. We tested more scenarios that assume partial trade restrictions, i.e. banning trade of polysilicon, ingots, wafers, cells, and modules respectively. According to the additional findings, we revised the explanation in Results and Discussion and added the scenario tests in Supplementary Information.

Similar effects can be expected if both the USA and Europe implement trade barriers against China – the unmet demand turns to India, Malaysia, Vietnam, and ROA (Supplementary Fig. 3). Banning products along the entire supply chain has greater impacts in rerouting supply chains than banning individual products (Supplementary Fig. 3). Moreover, banning the trade of downstream products simply shifts the dependence of upstream products on other regions, i.e., banning module imports leads to more imports of cells (Supplementary Fig. 3).

R3.7. The impact of geopolitical events (e.g., U.S.-China trade tensions) on supply chain resilience is not fully explored. Would alternative supply chains emerge in Southeast Asia or India under stricter trade restrictions?

Thank you for your inspiring comment. In agreement with comment R1.4, we test a scenario where the US and Europe both proceed trade restrictions against China across the whole PV supply chain, and added the new findings in Results, Discussion, and supporting figures in Supplementary Information.

Similar effects can be expected if both the USA and Europe implement trade barriers against China – the unmet demand turns to India, Malaysia, Vietnam, and ROA (Supplementary Fig. 3). Banning products along the entire supply chain has greater impacts in rerouting supply chains than banning individual products (Supplementary Fig. 3). Moreover, banning the trade of downstream products simply shifts the dependence of upstream products on other regions, i.e., banning module imports leads to more imports of cells (Supplementary Fig. 3).

R3.8. The manuscript should better address the limitations of the optimization approach. While it identifies cost-optimal solutions, real-world constraints such as political will, infrastructure availability, and investment risk should be discussed.

Thank you for your feedback. To better address the limitations of the optimization approach and highlight future opportunities to better consider real-world constraints including political will, infrastructure availability, and investment risk, we have extended the last paragraph of the manuscript:

However, our work has some limitations that ought to be addressed in future work. First, the model simplifies regional strategies to achieve a globally optimized solution, which

enables us to explore cost-and-job-optimal supply chain transitions under policy scenarios and assess trade-offs. However, it does not capture the full complexity of individual companies' actions or country-specific policies and our results should not be seen as projections. Real-world development is strongly influenced by political will¹⁴, infrastructure and capacity availability^{11,14}, and investment risk¹⁵. Future work could account for these factors by incorporating their consideration into the optimization process or by investigating likely development pathways^{16,17}.

R3.9. The sensitivity analysis is helpful but could be expanded. For example, how do variations in raw material costs (e.g., polysilicon price fluctuations) impact the scenario outcomes?

Thank you for your comment. We expanded the uncertainty analysis by testing the uncertainty caused by fluctuating product costs. We find that our results remain stable even in the extended analysis. Based on the new results, we extend our in-text discussion of the sensitivity analysis and we update Supplementary Figures 13-14.

Cost fluctuations in individual products do not significantly affect regional market shares; however, market shares are more sensitive to variations in downstream costs than upstream costs (Supplementary Fig. 13-14).

Supplementary Figure 3 Regional market shares by product for baseline scenario and maximize European jobs scenario, considering per-product the production cost fluctuations. The points show the median and the error bars show the range of 95% confidence interval. fluctuation of individual products does not make significant difference on the regional shares, although downstream products' cost fluctuations have bigger impacts on market share than do upstream products.

Supplementary Figure 4 Regional market shares by product for European module subsidies scenario and European trade barrier scenario, considering the per-product production cost fluctuations. . The points show the median and the error bars show the range of 95% confidence interval. Cost fluctuation of individual products does not make significant difference on the regional shares, although downstream products' cost fluctuations have bigger impacts on market share than do upstream products.

R3.10. The study mentions carbon emissions and job creation, but a deeper discussion on social and environmental trade-offs is needed.

Thank you for your suggestion. We deepened the discussion on the tradeoffs between social and environmental concerns in the Discussion.

Global supply chains also feature strong environmental and social trade-offs. For instance, localizing production in developed economies cuts emissions but reduces global jobs and opportunities for developing economies to benefit from low-carbon industry. Accounting for regional labor and environmental policies could further increase the global social and environmental trade-offs between local and international supply chains. For example, accounting for the EU's Carbon Border Adjustment Mechanism²⁴ would reduce the cost advantages of importing PV made in carbon-intensive processes and further strengthen the argument for localizing the PV industry. However, a more self-reliant Europe also reduces job opportunities and low-carbon technology development in Southeast Asia. Nonetheless, the trade-offs offered by globalized supply chains would change over time. Global cooperation could help reduce regional differences in labor intensities, and global decarbonization will reduce regional differences in the carbon intensity of manufacturing. Engaging in global cooperation paradoxically also increases regions' competitive positions as they acquire infrastructure, technological expertise, and a skilled workforce.

R3.11. Does the model account for environmental regulations affecting manufacturing locations (e.g., stricter labor and pollution controls in Europe vs. China)?

Thank you for your question. We do not represent specific environmental or labor policies within the model. However, the effect of different environmental policies is captured by the different considering different carbon intensities of production across manufacturing locations. However, the model does not capture environmental effects beyond carbon emissions nor does it account for environmental import controls, like the Carbon Border Adjustment Mechanism (CBAM). We explain these points in text at Supply Chain Parameters in Methods section:

***Emission factors** for production ($ProdEF_i^p$) and trade ($TradeEF_{i,j}^p$), are applied to the production⁶ and trade¹⁹ of each component. We focus on one example of social and environmental outputs—jobs and carbon emissions—and their variation between locations. Other impacts of real-world production, such as labor standards and pollution control, are beyond the scope of this study.*

We note that trade barriers may also be an extreme form of environmental and labor regulation. In the results, we explain how the manufacturing locations shift resulting from trade barriers:

Imposing a European trade barrier with China shifts the Europe's dependence from China to Malaysia, thus not totally achieving the goal of increased self-sufficiency but merely transferring dependency to another country. Similar effects can be expected if both the USA and Europe implement trade barriers against China – the unmet demand turns to India, Malaysia, Vietnam, and ROA (Supplementary Fig. 3).

Finally, we also comment on how accounting for labor and environmental controls would affect manufacturing location in the Discussion:

Accounting for regional labor and environmental policies could further increase the global social and environmental trade-offs between local and international supply chains. For example, accounting for the EU's Carbon Border Adjustment Mechanism²⁴ would reduce the cost advantages of importing PV made in carbon-intensive processes and further strengthen the argument for localizing the PV industry. However, a more self-reliant Europe also reduces job opportunities and low-carbon technology development in Southeast Asia.

R3.12. How do job creation estimates compare with actual labor market conditions in different regions?

Thank you for your comment. To provide the reader with this additional context, we compare our model results to the current labor market of PV manufacturing and discuss that point as local efforts to achieve globally diversified supply chain.

We compare the estimates with actual labor market conditions in the Discussion, where we write:

Achieving a global transition requires all regions to increase the size of their workforces, but regional efforts may differ considerably. Our results suggest that European and American policymakers need to increase their workforces by more than 120% each year to meet supply chain targets, while Asian regions require annual job growth below 30% (see Supplementary Table 2). Overall, localizing and maintaining PV supply chains will depend not only on investment, but also on rapidly expanding the available workforce.

R3.13. Figure 2 and Figure 3 contain valuable insights but require clearer labeling and captions.

Thank you for your comment. We revised the labeling and captions to avoid misunderstanding. Please see the revised figures below.

Fig. 4 Overview of global PV supply chains as shown by region-to-region trade flows for individual PV products. Outer rings compound production and demand for the ten indicated regions. Arrows indicate regional trade flows and show the share of supply and demand for each outer ring; for example, the black arrow in (a) emphasizes that the majority of China's polysilicon supply is used to meet the domestic national demand and the remaining demand is mainly supplied from Europe and Malaysia. (a) shows 2022 trade flows and (b) shows the 2030 trade flows according to the tested scenarios. Trade flows for individual products are shown left-to-right following the PV value chain, with modules being the products sold to end-consumers.

Fig. 5 The cost and impacts of European policy actions on PV supply chains. Panels (a-d) show the scenario-by-scenario results on global (a) cumulative costs, (b) cumulative emissions, (c) cumulative job creation, and (d) average self-sufficiency across years. Panels (e-g) show the cumulative cost-jobs trade-offs according to policy actions aiming to (e) maximize European jobs (MEJ), (f) maximize European jobs and provide a production subsidy, and (g) maximize European jobs and impose a trade barrier on all Chinese PV products. The point shading in panels (e-g) corresponds to different European self-sufficiency targets, which are modelled in the optimization model as a constraint.

R3.14. Some sentences, especially in the introduction and discussion, are overly long and complex. Consider simplifying key messages for better readability.

Thank you for your comment. Combining your comment and Reviewer #2's suggestions, we revised the Introduction. The discussion is also revised.